# Systemic Sclerosis-Associated ILD: Insights and Limitations of ScleroID

**DOI:** 10.3390/diagnostics16010158

**Published:** 2026-01-04

**Authors:** Cristina Niță, Laura Groșeanu

**Affiliations:** 1Faculty of Medicine, Carol Davila University of Medicine and Pharmacy, 050474 Bucharest, Romania; elec.nita@gmail.com; 2Sfanta Maria Clinical Hospital, 011172 Bucharest, Romania

**Keywords:** SSc-ILD, Disease activity, Patient-reported outcome

## Abstract

**Background/Objective:** Pulmonary involvement in systemic sclerosis (SSc) is typically assessed using pulmonary function tests (PFTs), high-resolution CT (HRCT), and composite indices. Patient-reported outcomes (PRO), including ScleroID, provide insight into quality of life, but their relationship with clinical measures and role in overall disease assessment remain unclear. To assess the correlation between ScleroID scores and both lung involvement and disease activity/damage in a cohort of SSc-ILD patients from a large tertiary care center. **Methods**: Disease activity [European Scleroderma Study Group Activity Index (EScSG-AI), Scleroderma Clinical Trials Consortium Activity Index (SCTC-AI)], disease severity [Medsger severity scale (MSS)], and PRO measure ScleroID were assessed for associations with the extent and severity of SSc-ILD. **Results:** In 82 patients with SSc-ILD (mean age 56.0 ± 10.8 years; median disease duration 4.2 ± 4.7 years), higher fibrosis extent (>20%) was associated with worse lung function, greater exercise limitation, and higher ScleroID scores, particularly in fatigue, social life, and body mobility domains (all *p* ≤ 0.03). Patients with >20% fibrosis also had worse NYHA class and Borg scores during 6-MWD (*p* < 0.001). Cross-sectional correlations showed that ScleroID total and individual domains were negatively associated with FVC% and 6-MWD, and positively with ILD extent on HRCT. Fatigue, social impact, and mobility domains correlated most strongly with disease activity and severity scores, especially in patients with > 20% fibrosis (r = 0.384–0.635, all *p* ≤ 0.016), whereas breathlessness showed minimal associations (r < 0.2). **Conclusions:** In SSc-ILD, greater lung fibrosis and functional impairment are associated with worse patient-reported quality of life, particularly in fatigue, mobility, and social domains. ScleroID scores reflect both physiological severity and disease burden highlighting its value as a multidimensional outcome measure in patients with more advanced disease.

## 1. Introduction

Systemic sclerosis (SSc) is a rare autoimmune disease marked by microvascular injury, immune activation, and progressive fibrosis of the skin and internal organs [1,2]. In SSc-associated interstitial lung disease (SSc-ILD), this triad drives lung injury through chronic endothelial damage, inflammation, and fibroblast activation, ultimately leading to irreversible extracellular matrix accumulation [3,4].

SSc-ILD is among the most severe complications, further impairing functional capacity and survival [5]. Recent expert consensus frameworks classify SSc ILD into subclinical, clinical, and progressive phenotypes, based on parameters such as forced vital capacity (FVC), gas exchange, and HRCT extent [6]. Management has evolved: immunosuppressants (e.g., mycophenolate mofetil), anti-fibrotic therapy (nintedanib), and biomarker-guided strategies are increasingly used in a personalized manner [7,8]. Moreover, emerging biomarkers (such as VCAM-1, SP-D, and CXCL4) show potential for predicting disease progression and outcomes in SSc ILD [9].

In contrast, the impact of SSc-ILD on quality of life (QoL) is less well characterized. Given the multisystemic nature of SSc, multiple organ manifestations contribute to overall disease burden, yet the point at which patients begin to experience meaningful functional decline remains insufficiently defined [10,11,12].

Accurate evaluation of disease burden requires integrating both clinical assessments and the patient’s perspective. Traditional physician-derived indices—such as the European Scleroderma Study Group Activity Index (EScSG-AI), the Scleroderma Clinical Trials Consortium Activity Index (SCTC-AI), and the Modified Severity Scale (MSS)—capture clinician-observed parameters but may not fully reflect the functional limitations and QoL impact experienced by patients [13,14,15].

In clinical studies of SSc-ILD, a variety of patient-reported outcome measures (PROMs) have been used to capture the impact of the disease and its treatments on QoL. Organ-specific instruments, such as the St. George’s Respiratory Questionnaire (SGRQ), King’s Brief Interstitial Lung Disease Questionnaire (K-BILD), and Functional Assessment of Chronic Illness Therapy–Dyspnea Scale (FACIT-D), primarily assess respiratory symptoms and their functional impact [16,17,18]. Generic PROMs, including the Short Form-36 Health Survey (SF-36), Health Assessment Questionnaire–Disability Index (HAQ-DI), and Patient-Reported Outcomes Measurement Information System-29 (PROMIS-29), evaluate overall health status and disability [19,20,21]. In the SENSCIS trial [22], PROMs included the SGRQ, FACIT-D, and HAQ-DI, incorporating the Scleroderma HAQ Visual Analogue Scale (SHAQ VAS) assessed at baseline and week 52 for associations with SSc-ILD severity [23]. Other commonly applied tools include the Leicester Cough Questionnaire (LCQ), Mahler’s Dyspnea Index (MDI), and Baseline and Transition Dyspnea Indices (BDI/TDI) [24,25,26]. Recent studies, such as a prospective investigation correlating QoL with disease parameters and the Phase 2 ATHENA-SSc-ILD trial evaluating PRA023, have also integrated PROMs to assess patient-centered outcomes and the impact of therapeutic interventions on QoL [27,28].

Building on previous work, a novel PROM, the EULAR Systemic Sclerosis Impact of Disease (ScleroID), was recently introduced [29,30]. The ScleroID is a brief, patient-derived questionnaire designed to capture self-assessed disease severity in SSc. It comprises 10 items across multiple domains, including physical function and organ involvement, with two items on fatigue and respiratory difficulty, which may be particularly relevant in patients with pulmonary involvement. ScleroID has been validated across multiple European centers, including Romania, showing strong internal consistency, high reliability (intraclass correlation coefficient = 0.839), and sensitivity to change over time. Its role capturing the impact of SSc-ILD on patients’ daily functioning and QoL remains unclear, prompting the present investigation into its relationship with lung function. This reinforces the importance of integrating objective measures of organ involvement with patient-reported outcomes, as recommended in recent EULAR and ATS guidelines for comprehensive SSc assessment [31,32].

## 2. Materials and Methods

### 2.1. Study Design

This prospective observational study was conducted from 15 October 2023 to 30 August 2025. All participants met the 2013 American College of Rheumatology/European Alliance of Associations for Rheumatology (ACR/EULAR) classification criteria for SSc [33]. All individuals presenting to the clinic during the study period were systematically screened, and those with high-resolution computed tomography (HRCT) of the thorax—confirmed SScILD were enrolled consecutively to reduce selection bias. Additional inclusion criteria were age ≥ 18 years, availability of baseline pulmonary function tests (PFTs), and HRCT within 6 months prior to enrolment. Exclusion criteria included the absence of ILD on HRCT, overlap connective tissue diseases, alternative causes of ILD, and incomplete baseline data.

We retrieved pulmonary function tests (PFTs) and electronic lung HRCT image files from both baseline and the last available follow-up visit. The extent of lung fibrosis on HRCT, characterized by the presence of reticular changes and/or honeycombing, was classified as either <20% or ≥20% relative to the total lung volume [34]. PFTs including diffusing lung capacity for carbon monoxide (DLCO), forced vital capacity (FVC), and forced expiratory volume during the first second (FEV1) were conducted following the guidelines of the American Thoracic Society/European Respiratory Society (ERS) [35]. We also recorded the 6 min walk distance test (6-MWD) and assessed dyspnea symptoms using functional classes [36]. Documentation of right heart catheterization (RHC) was noted when performed, and pulmonary hypertension (PH) was diagnosed according to the 2015 European Society of Cardiology/ERS guidelines, defining PH as a mean pulmonary arterial pressure (mPAP) ≥25 mmHg, measured with RHC [37]. In the absence of RHC, PH was defined as a systolic arterial pressure (sPAP) >40 mmHg on the echocardiography. Patients with available data over a 10-year follow-up period were evaluated for ILD progression, which was assessed by absolute changes in percentage predicted from baseline to follow-up, and defined as severe (total FVC decline >10%), moderate (FVC decline, 5–10%), or stable FVC (≤5% change) [38,39,40].

The demographic and clinical characteristics of our cohort—including age, sex distribution, disease duration, autoantibody profile, extent of ILD on HRCT, and baseline PFTs—are consistent with those reported in large SSc-ILD registries and trials, such as the EUSTAR database [41], progressive ILD analyses [42], and real-life two-centre cohorts [35]. These similarities support the representativeness of our sample and the generalizability of our findings.

The study aimed to evaluate the associations between the ScleroID, disease activity, and severity scores with baseline lung involvement in SSc-ILD patients. Disease activity was assessed using the European Scleroderma Study Group Activity Index (EScSG-AI) and the Scleroderma Clinical Trials Consortium Activity Index (SCTC-AI), both of which combine clinical and laboratory domains, with higher indicating greater disease activity. Disease severity was assessed using the Medsger Severity Scale (MSS), which rates major organ involvement on a 0–4 scale (0 = no involvement, 4 = severe involvement). QoL was measured using the self-administered validated Romanian version of the ScleroID questionnaire, in addition to the clinical indices above [43].

### 2.2. Statistics

Baseline characteristics of the cohort were summarized using descriptive statistics, and the distribution of continuous variables was assessed with the Shapiro–Wilk test. Group comparisons were performed using two-sample *t*-tests, Chi-square tests, Kruskal–Wallis tests, or Mann–Whitney U tests, as appropriate. Relationships between patient-reported outcomes, disease activity and severity indices, and lung-specific measures were explored using Spearman’s rank correlation coefficient (rho). Patients were stratified by HRCT-assessed lung fibrosis extent (10–20% vs. ≥20%) to determine whether associations differed by fibrosis severity. All statistical tests were two-sided, and a *p*-value < 0.05 was considered statistically significant. Analyses were conducted using IBM SPSS Statistics version 31.0.0.0, and graphical representations of correlations and group comparisons were generated to visualize trends in disease activity, severity, and patient-reported impact across fibrosis strata.

## 3. Results

### 3.1. Baseline Characteristics

Mean age was 56.0 (±10.8) years and median time since first non-Raynaud’s symptom was 4.2 (±4.7) years. Mean FVC% predicted was 76.8%, and mean DLCO% predicted was 54.3%. Mean baseline EscSG-AI total score was 6.1 (±1.7), baseline SCTC-AI was 34.5 (±14.8), Medsger severity score was 9.6 (±3.8), ScleroID total 4.1 (±2.4), and Breathlessness item of ScleroID score was 3.8 (±2.9). While SCTC-AI and EscSG-AI socres did not differ by sex, age, or disease duration, patients with diffuse cutaneous SSc had higher SCTC-AI scores (*p* = 0.03), and ATA positivity was associated with higher EscSG-AI scores (*p* = 0.04). Additionally, dyspnea of NYHA class > III and FVC < 80% predicted correlated with higher EScSG-AI scores (*p* = 0.05 and *p* = 0.04, respectively), and notably, HRCT fibrosis extent >20% and the presence of pulmonary hypertension were both linked to significantly higher scores across all instruments (*p* < 0.001) (Table 1).

Patients with ≥20% fibrosis on HRCT at baseline reported worse mean scores in most ScleroID items than those between 10–20%, again reflecting worse QoL in patients with more severe disease: Fatigue 5.56 vs. 3.97 (*p* = 0.02), Social life 5.23 vs. 2.88 (*p* = 0.001), Body mobility 5.10 vs. 3.32 (*p* = 0.01), Breathlessness 4.64 vs. 2.61 (*p* = 0.003). Similarly, NYHA functional class and worst Borg scale during the 6-MWD reported worse mean scores (Figure 1).

### 3.2. Cross-Sectional Associations Between ScleroID Items and Baseline Lung Parameters

Higher ScleroID scores were associated with worse lung function and reduced exercise capacity, as reflected by lower FVC% and shorter 6 min walk distance. Both total and individual ScleroID items correlated positively with ILD extent on HRCT (Figure 2). To explore whether these associations were preserved or varied across different levels of lung involvement, we stratified patients by fibrosis extent. After stratification, the expected interrelationship between ScleroID domains (fatigue, social impact, mobility, and breathlessness), were largely maintained in both the 10–20% and >20% fibrosis groups. Fatigue correlated with social impact (r = 0.700 vs. 0.686, *p* < 0.001), mobility (r = 0.618 vs. 0.778, *p* < 0.001), and breathlessness (r = 0.598 vs. 0.737, *p* < 0.001), which in turn was also associated with reduced mobility (r = 0.772 vs. 0.588, *p* < 0.001). Notable subgroup-specific differences emerged, particularly when examining objective exercise-related measures. In the 10–20% fibrosis group, perceived breathlessness and reduced mobility were strongly associated with exercise-induced limitations, such as desaturation (r = −0.561 and −0.602) and higher Borg scores (r = 0.641 and 0.531), whereas these correlations were generally weaker in patients with >20% fibrosis. NYHA class correlations were similar for fatigue in both fibrosis categories (r = 0.466), but in patients with >20% fibrosis, NYHA class was also strongly associated with social impact (r = 0.608, *p* < 0.001) and reduced mobility (r = 0.568, *p* < 0.001), underscoring the greater functional and social burden in advanced disease.

### 3.3. Impact of SSc Disease Activity and Damage on QoL in the Studied SSc-ILD Cohort

Regarding the correlation between disease activity, severity, and ScleroID, fatigue, social impact, and mobility were the domains most related to activity and severity scores. The breathlessness domain showed minimal associations with disease activity or severity in either fibrosis subgroups (all r < 0.2). By contrast, fatigue, social impact, and mobility were the ScleroID domains most closely related to disease burden. As expected, these associations were strongest in patients with >20% fibrosis, where fatigue and mobility correlated with overall disease scores (r = 0.384, *p* = 0.016; r = 0.525, *p* < 0.001), and all three domains aligned with SCTC total scores (r = 0.515–0.635, all *p* < 0.001). In the 10–20% fibrosis groups, correlations were generally weaker, with only minor associations observed for fatigue [(r = −0.341, *p* = 0.045) with SCTC total; (r = −0.359, *p* = 0.034) with EScSG AI] (Figure 3).

## 4. Discussion

In this study, we evaluated the associations between disease severity, functional parameters, and PROMs using the ScleroID questionnaire in patients with SSc-ILD. Our findings demonstrate that objective measures of disease burden and patient-perceived limitations are closely intertwined, yet each provides complementary information.

Functional impairments, such as NYHA class > III dyspnea and FVC < 80% predicted, were associated with higher ESsSG-AI scores, underscoring the influence of pulmonary function on both clinician-assessed and patient-reported disease activity. Interestingly, even in patients with significant pulmonary involvement—such as those with FVC below 80% predicted or experiencing a decline greater than 10%—the mean ScleroID score remained relatively low (4.8 ± 2.3). This suggests that factors beyond objective pulmonary function contribute to patient-reported outcomes in SSc. Paradoxically, patients with less severe dyspnea (NYHA class < III) sometimes reported higher ScleroID scores, highlighting the complex interplay between symptoms, perception, and disease burden.

Patients with more severe ILD (HRCT fibrosis > 20%, NYHA class > III, reduced FVC, or pulmonary hypertension) reported worse ScleroID scores across the fatigue, social impact, mobility, and breathlessness domains. This reinforces that PROMs are sensitive to objective measures of disease severity and functional impairment, and highlights the utility of ScleroID as a complement to standard clinical and radiographic assessments.

Prior studies have similarly shown that ILD extent and functional impairment correlate with worse HRQoL in SSc. For example, a 2025 ILD PRO registry analysis demonstrated that HRCT fibrosis extent predicts prognosis more strongly that pattern alone [45], and a post hoc SENSCIS trial analysis found that lower FVC (<70% predicted) and greater fibrosis (>30%) were associated with worse PROMs (SGRQ, FACIT Dyspnoea, HAQ DI, SHAQ VAS), with 1-year FVC changes paralleling shifts in PROMs [46].

Subgroup analyses based on HRCT fibrosis extent revealed that in patients with 10–20% fibrosis, perceived breathlessness and reduced mobility closely reflected exercise-induced limitations, including desaturation and higher Borg scores, whereas these associations were weaker in >20% fibrosis. This suggests that as fibrosis progresses, patients’ perceptions may no longer fully capture functional impairment, likely due to the interplay of cardiopulmonary limitations beyond parenchymal fibrosis and compensatory mechanisms [47]. These observations align with evidence that HRCT extent correlates with outcomes but does not entirely explain functional decline [48].

Fatigue emerged as a significant correlate of disease activity in the >20% fibrosis subgroup, consistent with prior studies highlighting fatigue as a prevalent and debilitating symptom in SSc, significantly affecting patients’ quality of life and social participation [49]. Mobility impairment also correlated with disease severity, reflecting contractures and muscle weakness in advanced SSc-ILD. The robust relationship between mobility and disease severity in advanced fibrosis stages highlights the utility of mobility assessments in monitoring disease progression [50]. The social impact domain correlated with total disease burden underscoring how physical limitations translate into psychosocial consequences. In contrast, in patients with 10–20% fibrosis, fatigue and breathlessness showed weak correlations with disease activity, likely reflecting limited pulmonary involvement. Clinically, this suggests that in this subgroup, PROMs alone may underestimate disease burden, and should therefore be complemented with objective assessments. This is suported by previous studies and expert consensus indicating that patient-reported symptoms may not fully allign with radiographic fibrosis, particularly in early SSc-ILD [27,48,51,52]. For instance, a prospective study of 63 SSc ILD patients found significant correlations between lung function (FVC and DLCO) and patient-reported dyspnea (measured with the FACIT dyspnea questionnaire), but baseline and one year HRCT scores did not strongly correlate with quality-of-life measures [27]. Similarly, a recent study reported that a substantial proportion of ILD patients were asymptomatic at diagnosis, despite HRCT-detected fibrotic changes [52].

This study has several limitations. First, its cross-sectional design precludes causal inferences between ScleroID scores and disease progression. Longitudinal studies with longer follow-up are required to confirm these results and explore trajectories of change in disease burden. Second, the relatively small fibrosis-based subgroups limit statistical power and highlight the need for larger, multi-center cohorts to validate subgroup-specific associations. Third, as a PROM, ScleroID may be influenced by psychological, social, or comorbid factors not assessed here, introducing potential reporting biases. Finally, not all relevant clinical manifestations or functional parameters (e.g., high-resolution imaging metrics or specific inflammatory biomarkers) were included, which may limit the comprehensiveness of correlations between ScleroID scores and disease activity and severity.

## 5. Conclusions

ScleroID captures patient-perceived disease burden in SSc-ILD, complementing traditional clinical and imaging assessments. In advanced fibrosis, scores closely reflect objective disease severity, supporting their use in monitoring progression and guiding interventions. In early fibrosis, PROs reveal subtle limitations that may be overlooked, emphasizing their value for proactive, individualized management. These correlations between ScleroID scores and clinical measures can guide personalized patient management and enhance disease monitoring strategies, ensuring a more patient-centered approach to care.

## Figures and Tables

**Figure 1 diagnostics-16-00158-f001:**
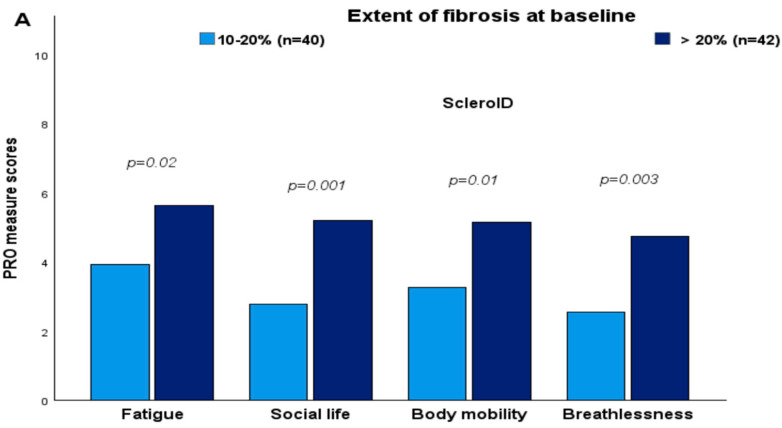
Higher HRCT fibrosis extent was associated with worse ScleroID scores and functional impairment (**A**) ScleroID scores in patients with 10–20% vs. >20% lung fibrosis (**B**) NYHA class and Borg scores during 6-MWD in patients with 10–20% vs. 20% fibrosis. Statistical significance was assessed using an unpaired Student’s *t*-test for two groups.

**Figure 2 diagnostics-16-00158-f002:**
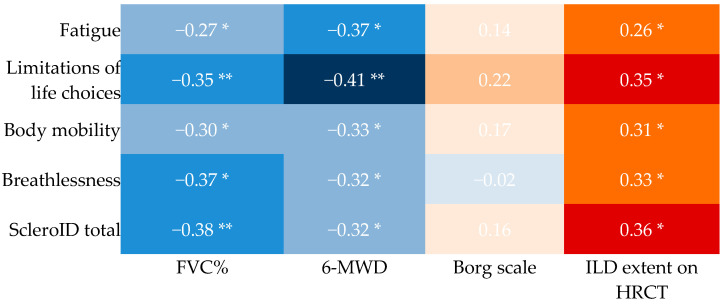
Cross-sectional associations between ScleroID items and baseline lung parameters. Each cell displays the Spearman rho correlation coefficient with its significance (* *p* < 0.05, ** *p* < 0.01). Blue shading indicates negative correlations (rho < 0), and red shadowing indicates positive correlations (rho > 0). Color intensity corresponds to magnitude of the correlation (darker shades = stronger correlations). FVC: forced vital capacity; 6-MWD: 6 min walk distance; ILD: interstitial lung disease; HRCT: high-resolution CT.

**Figure 3 diagnostics-16-00158-f003:**
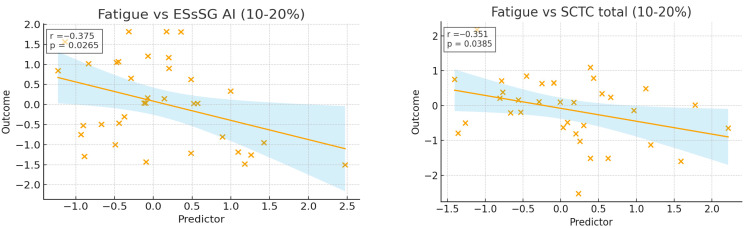
Relationship Between ScleroID Domains (Fatigue, Mobility, Social Impact) and Disease Activity/Severity Scores Stratified by Skin Fibrosis Extent Fatigue, mobility, and social impact were most strongly associated with disease activity and severity, particularly in >20% fibrosis, while breathlessness showed minimal correlations.

**Table 1 diagnostics-16-00158-t001:** Baseline ScleroID and related outcome measures.

	Baseline Mean (SD) Composite Measures			
Baseline Characteristics	EscSG-AI	SCTC-AI	MSS	ScleroID
Gender				
Female (*n* = 68)	6.1 (2.2)	34.0 (15.6)	8.9 (3.9)	4.0 (2.2)
Male (*n* = 14)	5.6 (1.8)	34.5 (15.8)	9.7 (3.1)	4.1 (2.6)
Age				
<65 years (*n* = 59)	6.0 (2.2)	33.5 (14.8)	9.7 (3.8)	4.1 (2.3)
≥65 years (*n* = 23)	6.2 (2.1)	37.7 (17.9)	9.3 (3.8)	3.8 (2.2)
SSc subset				
dcSSc (*n* = 44)	6.2 (2.1)	37.1 (15.8)	10.2 (4.1)	4.5 (2.4)
lcSSc(*n* = 38)	5.8 (2.0)	30.8 (13.9)	8.9 (3.3)	3.6 (2.1)
SSc disease duration ^a^				
≤3 years (*n* = 43)	5.8 (1.8)	32.1 (13.2)	9.1 (3.7)	4.0 (2.3)
>3 years (*n* = 39)	6.3 (1.6)	36.9 (16.0)	10.2 (3.8)	4.2 (2.4)
Autoantibodies				
Anti-centromere (*n* = 14)	5.5 (1.9)	32.3 (13.6)	9.4 (3.8)	4.5 (2.3)
Anti-toposiomerase (*n* = 54)	6.4 (1.5)	35.6 (15.3)	9.8 (3.8)	3.8 (2.3)
6-MWD desaturation < 94% OR ≥ 5% ^b^				
Yes (*n* = 21)	5.6 (1.9)	35.2 (14.3)	9.2 (3.7)	4.2 (2.3)
No (*n* = 22)	6.2 (1.6)	32.5 (14.5)	9.7 (3.9)	3.9 (2.5)
Unexplained dyspnea functional class 3 or 4				
Yes (*n* = 31)	5.3 (1.9)	30.9 (12.3)	8.4 (3.8)	3.6 (2.0)
No (*n* = 51)	6.3 (1.6)	34.5 (15.0)	9.8 (3.7)	4.0 (2.6)
FVC % predicted at baseline				
<80% (*n* = 53)	5.6 (1.9)	33.1 (14.1)	8.9 (3.8)	3.5 (2.3)
≥80% (*n* = 29)	6.5 (1.4)	34.1 (16.4)	10.4 (4.0)	4.0 (2.3)
>10% FVC decline on follow-up PFT				
Yes (*n* = 45)	6.3 (1.6)	35.1 (15.9)	10.1 (4.1)	4.3 (2.5)
No (*n* = 37)	5.7 (1.8)	33 (13.4)	9.0 (3.5)	3.7 (2.1)
Extent of fibrosis by HRCT				
10–20% (*n* = 40)	5.5 (1.7)	28.8 (12.5)	7.4 (2.6)	3.1 (2.1)
≥20% (*n* = 42)	6.7 (1.4)	40.2 (14.6)	11.6 (3.7)	4.8 (2.3)
PH				
Yes (*n* = 29)	6.4 (1.7)	44.2 (13.8)	12.1 (3.8)	4.8 (2.1)
No (*n* = 53)	5.9 (1.7)	29.1 (12.4)	8.3 (3.1)	3.7 (2.4)

Data are presented as median ± interquartile range or *n* (%), unless otherwise stated. SSc: systemic sclerosis; dcSSc: difusse cutaneous SSc; lcSSc: limited cutaneous SSc; FVC: forced vital capacity; HRCT: high-resolution computed tomography; PH: pulmonary hypertension. ^a^ disease duration: from first non-Raynaud’s symptom to baseline visit; ^b^ 6 min walk distance desaturation after the test [44]; *p*-values of univariate comparisons of baseline characteristics between the two cohorts are not shown. Mann–Whitney U-test was used to compare continuous variables.

## Data Availability

The original contributions presented in this study are included in the article. Further inquiries can be directed to the corresponding author.

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
