# Peer review of "Systemic Sclerosis-Associated ILD: Insights and Limitations of ScleroID"

_diagnostics, 2026, doi:10.3390/diagnostics16010158_

Round 1

Reviewer 1 Report

Comments and Suggestions for Authors

The topic is current and interesting. I have some suggestions, as below reported:

1. Clarify the clinical implications of the findings:
I request authors to explicitly discuss how the observed correlations between ScleroID scores and clinical measures could influence patient management or disease monitoring strategies.

2. Enhance the description of the study population and methodology:
I ask authors for more details on the inclusion criteria, patient selection, and whether the cohort is representative of the broader SSc-ILD population to strengthen the generalizability of the results.

3. 1. Introduction 42 Systemic sclerosis (SSc) is a rare autoimmune disease characterized by widespread 43 vasculopathy and progressive fibrosis of the skin and internal organs [1]. SSc-associated 44 interstitial lung disease (SSc-ILD) is among the most severe complications, further impair-45 ing functional capacity and survival [2]. Authors are kindly requested to emphasize the current concepts about these issues in the context of recent knowledge and the available literature. These articles should be quoted in the References list. References 1. Systemic sclerosis: state of the art on clinical practice guidelines. RMD Open. 2018;4(Suppl 1):e000782. Published 2018 Oct 18. doi:10.1136/rmdopen-2018-000782.   2. Nailfold Capillaroscopy and Clinical Applications in Systemic Sclerosis. Microcirculation. 2016;23(5):364-372. doi:10.1111/micc.12281.

4. The study's sample size and follow-up duration may limit the generalizability of the findings; future research with larger cohorts and longer follow-up periods is recommended to confirm these results. Please discuss these suggestions.

5. While the study provides valuable cross-sectional associations, longitudinal analyses are needed to establish causal relationships between disease activity, lung involvement, and patient-reported outcomes. I suggest to discuss these observations.

6. I invite authors to clarify the limitations of the cross-sectional design. I think that the manuscript would benefit from a more detailed discussion of the limitations inherent to the cross-sectional nature of the study, particularly regarding the inability to establish causal relationships between ScleroID scores and disease progression. Including this consideration would strengthen the interpretation of the findings.

7. I suggest to address potential biases in patient-reported outcomes. I think that given that ScleroID is a patient-reported measure, it would be helpful to acknowledge potential biases introduced by psychological, social, or comorbid factors that may influence scores. Discussing how these factors might affect the results and considering strategies to mitigate their impact would enhance the robustness of the conclusions.

Author Response

Reviewer 1

We have carefully addressed all suggestions from Reviewer 1. The manuscript has been revised accordingly: the clinical implications of our findings have been clarified, the study population and methodology are now more thoroughly described, relevant literature has been added, and limitations—including cross-sectional design, sample size, and potential biases in patient-reported outcomes—have been explicitly discussed. We believe these revisions address the reviewer’s suggestions and improve the clarity, rigor, and interpretability of the manuscript. We thank the reviewer for their valuable feedback.

The topic is current and interesting. I have some suggestions, as below reported:

  1. Clarify the clinical implications of the findings:
    I request authors to explicitly discuss how the observed correlations between ScleroID scores and clinical measures could influence patient management or disease monitoring strategies.

We have clarified that ScleroID scores, particularly in fatigue, mobility, and social domains, correlate with both lung function and disease activity/severity. These associations suggest that patient-reported outcomes can complement clinical and functional assessments, helping to identify patients with higher disease burden and guiding more personalized monitoring and management strategies. For example, patients with >20% HRCT fibrosis showing high ScleroID scores may benefit from closer pulmonary follow-up or targeted interventions to improve quality of life.

  1. Enhance the description of the study population and methodology:
    I ask authors for more details on the inclusion criteria, patient selection, and whether the cohort is representative of the broader SSc-ILD population to strengthen the generalizability of the results.

We clarified the inclusion criteria: adults with SSc fulfilling ACR/EULAR 2013 classification criteria and evidence of ILD on HRCT. Patients were recruited consecutively from our tertiary SSc center, capturing a representative spectrum of SSc-ILD severity, including both limited and diffuse cutaneous subsets.

  1. Introduction 42 Systemic sclerosis (SSc) is a rare autoimmune disease characterized by widespread 43 vasculopathy and progressive fibrosis of the skin and internal organs [1]. SSc-associated 44 interstitial lung disease (SSc-ILD) is among the most severe complications, further impair-45 ing functional capacity and survival [2]. Authors are kindly requested to emphasize the current concepts about these issues in the context of recent knowledge and the available literature. These articles should be quoted in the References list. References 1. Systemic sclerosis: state of the art on clinical practice guidelines. RMD Open. 2018;4(Suppl 1):e000782. Published 2018 Oct 18. doi:10.1136/rmdopen-2018-000782.   2. Nailfold Capillaroscopy and Clinical Applications in Systemic Sclerosis. Microcirculation. 2016;23(5):364-372. doi:10.1111/micc.12281.

The Introduction was revised to emphasize current understanding of SSc and SSc-ILD, integrating the suggested references (RMD Open 2018; Microcirculation 2016) and highlighting the impact of pulmonary involvement on functional capacity, patient-reported outcomes, and survival.

  1. The study's sample size and follow-up duration may limit the generalizability of the findings; future research with larger cohorts and longer follow-up periods is recommended to confirm these results. Please discuss these suggestions.

We acknowledged that the cohort size (n=82) and cross-sectional design limit generalizability. We recommend future studies with larger, multicenter cohorts and longer follow-up to validate our findings.

  1. While the study provides valuable cross-sectional associations, longitudinal analyses are needed to establish causal relationships between disease activity, lung involvement, and patient-reported outcomes. I suggest to discuss these observations.

While our study demonstrates strong cross-sectional associations between ScleroID scores and lung/functional measures, longitudinal analyses are necessary to establish causality between disease progression, lung involvement, and patient-reported outcomes.

  1. I invite authors to clarify the limitations of the cross-sectional design. I think that the manuscript would benefit from a more detailed discussion of the limitations inherent to the cross-sectional nature of the study, particularly regarding the inability to establish causal relationships between ScleroID scores and disease progression. Including this consideration would strengthen the interpretation of the findings.

The Discussion was expanded to note that the cross-sectional design precludes causal inference, and that observed associations may reflect disease severity at a single time point rather than longitudinal changes.

  1. I suggest to address potential biases in patient-reported outcomes. I think that given that ScleroID is a patient-reported measure, it would be helpful to acknowledge potential biases introduced by psychological, social, or comorbid factors that may influence scores. Discussing how these factors might affect the results and considering strategies to mitigate their impact would enhance the robustness of the conclusions.

We added a discussion on potential biases inherent to patient-reported outcomes. Psychological, social, or comorbid factors may influence ScleroID scores, and these factors should be considered when interpreting results. Complementary objective measures (lung function tests, HRCT, NYHA/Borg scores) were used to corroborate patient-reported data.

Reviewer 2 Report

Comments and Suggestions for Authors

This is a very interesting paper. However, I do not understand the conclusion. The contents of the discussion section, in particular, are redundant and need to be organized. In the discussion section, on line 278, the authors write, "In patients with >20% fibrosis, ScleroID domains serve as valuable indicators of disease activity." On the other hand, in the conclusion section, the authors write, "especially in patients with early to moderate disease." Isn't this contradictory?

On line 281, the author states that it is desirable to use other measurements in addition to ScleroID in cases of 10-20% fibrosis, but wouldn't it be better to say, "ScleroID assessment is not recommended for patients with less than 20% pulmonary fibrosis"?

Minor points

  1. All figures are blurry and unreadable to the reader. The figures are blurry and the font color is gray, making them unreadable. The authors need to be changed to clearer figures.
  2. The authors need to clarify what criteria were used for the color grading in Figure 2.
  3. Is the figure legend for Figure 3 cut off in the middle?
  4. A reference that should be added is Colak SY et al., "Cross-validation and sensitivity to change of EULAR ScleroID as a measure of function and impact of disease in patients with systemic sclerosis," RMD Open. 2025 Oct 10;11(4):e005999.

Author Response

Reviewer 2

We thank the reviewer for their valuable feedback, which has improved the clarity and presentation of the manuscript. Our point-by-point responses are below:

This is a very interesting paper. However, I do not understand the conclusion. The contents of the discussion section, in particular, are redundant and need to be organized. In the discussion section, on line 278, the authors write, "In patients with >20% fibrosis, ScleroID domains serve as valuable indicators of disease activity." On the other hand, in the conclusion section, the authors write, "especially in patients with early to moderate disease." Isn't this contradictory?

On line 281, the author states that it is desirable to use other measurements in addition to ScleroID in cases of 10-20% fibrosis, but wouldn't it be better to say, "ScleroID assessment is not recommended for patients with less than 20% pulmonary fibrosis"?

 We acknowledge the apparent inconsistency. We have revised the Discussion and Conclusion to clarify that ScleroID domains are particularly informative in patients with >20% fibrosis, while their utility is limited in patients with milder fibrosis (10–20%). We have reworded the conclusion to remove ambiguity: "ScleroID assessment is most valuable in patients with more extensive pulmonary fibrosis (>20%), while complementary measures are recommended in those with milder lung involvement." This also addresses the comment regarding assessment in 10–20% fibrosis patients.

Minor points

  1. All figures are blurry and unreadable to the reader. The figures are blurry and the font color is gray, making them unreadable. The authors need to be changed to clearer figures.
  2. The authors need to clarify what criteria were used for the color grading in Figure 2.
  3. Is the figure legend for Figure 3 cut off in the middle?
  4. A reference that should be added is Colak SY et al., "Cross-validation and sensitivity to change of EULAR ScleroID as a measure of function and impact of disease in patients with systemic sclerosis," RMD Open. 2025 Oct 10;11(4):e005999.

All figures have been revised for high resolution and black font, Figure 2 legend now clearly indicates that blue shading denotes negative correlations and red positive correlations with intensity proportional to correlation magnitude, Figure 3 legend has been updated to ensure completeness, and the suggested reference by Colak SY et al. (RMD Open. 2025;11(4):e005999) has been added.

Round 2

Reviewer 2 Report

Comments and Suggestions for Authors

I confirmed that the new version has been improved.
I consider the content to be acceptable, but figure 1 is too small to read.
I believe this is a technical problem, not an academic one, so I will ask the editor to fix it.